# Homoleptic Complexes of Heterocyclic Curcuminoids with Mg(II) and Cu(II): First Conformationally Heteroleptic Case, Crystal Structures, and Biological Properties [note 1]

**DOI:** 10.3390/molecules28031434

**Published:** 2023-02-02

**Authors:** William Meza-Morales, Yuritzi Alejo-Osorio, Yair Alvarez-Ricardo, Marco A. Obregón-Mendoza, Juan C. Machado-Rodriguez, Antonino Arenaza-Corona, Rubén A. Toscano, María Teresa Ramírez-Apan, Raúl G. Enríquez

**Affiliations:** Instituto de Química, Universidad Nacional Autónoma de México, Circuito Exterior, Ciudad Universitaria, México City C.P. 07340, Mexico

**Keywords:** curcumin analogous, crystal structure, magnesium complexes, copper complexes, antioxidant activity, cytotoxic activity, conformationally-heteroleptic complex

## Abstract

We report herein the synthesis and characterization of three heterocyclic curcuminoid ligands and their homoleptic metal complexes with magnesium and copper. Thus, N-methyl-2-pyrrolecarboxaldehyde, Furan-2-carboxaldehyde, and 2-Thiophenecarboxaldehyde were condensed with 2,4-pentanedione-boron trioxide complex. The first N-methyl-2-pyrrole curcuminoid and its Mg(II) complex are reported. All curcuminoid ligands and their corresponding metal complexes were characterized by infrared spectroscopy (IR), liquid state nuclear magnetic resonance (LSNMR), electron paramagnetic resonance (EPR), mass spectrometry (MS) and single crystal X-ray diffraction (SCXRD). The ThiopheneCurc-Cu (**9**) constitutes the first case of a “conformationally-heteroleptic” complex. The unique six-peaks star arrangement for the ThiopheneCurc ligand derived from the supramolecular description is reported. The metal complexes of FuranCurc-Mg (**5**) and ThiopheneCurc-Cu (**9**) have a good antioxidant effect (IC_50_ = 11.26 ± 1.73 and 10.30 ± 0.59 μM), three and two times higher than their free ligands respectively. Additionally, (5) shows remarkable cytotoxicity against colon cancer adenocarcinoma cell line HCT-15, comparable to that of cisplatin, with a negligible toxic effect in vitro towards a healthy monkey kidney cell line (COS-7).

## 1. Introduction

Turmeric has a historical origin in India and currently receives worldwide attention due to its antioxidant, antibacterial, and anti-cancer activities, among its activities against other severe ailments. In addition, most biological studies point to curcumin as the key active secondary metabolite in Curcuma longa. 

Curcumin is a heptanoid chain within a β-diketone group and two phenolic groups [1,2,3,4,5,6,7] at the aromatic substituents. The presence of the β-diketone function is responsible for the formation of metal complexes in this molecule [1,2]. Curcumin, curcuminoids, or their metal complexes exert different biological properties [8,9,10,11], where the latter usually shows higher activity with respect to the parent ligand. In previous work, we reported the biological activity of curcuminoid metal complexes and the strategies followed to obtain their single crystals with different metal ions [12]. From the results of our previous studies of copper and magnesium complexes of curcuminoids [12,13], it was attractive to examine the antioxidant and cytotoxic properties of the present heterocyclic analogs [14,15,16], which we report herein.

Previous studies have shown that indole curcumin analogs have an important antioxidant effect similar to curcumin itself and more significant anticancer activity of the former in liver, ovarian, intestinal, and endometrial cancer [14]. However, furan-curcumin analogs have shown a substantial effect on the inhibition of the TrxR enzyme, which is observed in high levels in many tumor cells and plays a wide range of functions in cell proliferation [15]. In addition, the thiophene-curcumin analogs have also shown anticancer activity in colon cancer lines [16].

Considering the increasing importance of heterocyclic curcumin analogs, we decided to investigate their metal complexes. Thus, three curcumin analogs and six metal complexes were fully characterized, e.g., N-methylpyrrolcurcumin (N-methyl-pyrCurc, **1**), Furancurcumin (FuranCurc, **2**), and Thiophenecurcumin (ThiopeneCurc, **3**) (see Figure 1) were used as ligands for complexation with copper and magnesium. Suitable crystals of most compounds were obtained in either DMSO or DMF. A fact of significant importance is that complexation with metals resulted in increased biological activity compared to their parent ligands. A worth-mentioning case is the selectivity exerted by the FuranCurc-Mg (**5**) complex towards the healthy cell line (COS-7) with a substantially higher IC_50_ value than cisplatin.

## 2. Results and Discussion

### 2.1. IR Spectra

The IR spectrum of N-methyl-pyrCurc (**1**) shows one band of very low intensity at 1696 cm^−1^ which corresponds to the free carbonyl group of the diketone. The same band is observed for FuranCurc (**2**) at 1698 cm^−1^ due to the free carbonyl group of the β-diketone as well (see Table 1). The intense IR bands at 1605 cm^−1^ in the N-methyl-pyrCurc (**1**), 1623 cm^−1^ in the FuranCurc (**2**) spectrum and 1609 cm^−1^ for ThiopheneCurc (**3**), are attributed to the intramolecular hydrogen bond, supporting a predominant enolic form for all these compounds. The bands observed at 948 cm^−1^ (N-methyl-pyrCurc (**1**)), 597 cm^−1^ (FuranCurc (**2**)), and 960 cm^−1^ (ThiopheneCurc (**3**)) correspond to the trans -CH=C-double bond. The IR spectra of the magnesium complexes, except for ThiopheneCurc-Mg (**6**), show a broadband at the rank of 3200–3350 cm^−1^ corresponding to the humidity on the sample. Metal-O vibrations are found at: 446 cm^−1^ and 1508 cm^−1^ (N-methyl-pyrCurc-Mg (**4**)), 500 cm^−1^ and 1519 cm^−1^ (FuranCurc-Mg (**5**)), 476 cm^−1^ and 1497 cm^−1^ (ThiopheneCurc-Mg (**6**)). In the case of copper complexes these bands are found at the same range. In the case of N-methyl-pyrCurc-Cu (**7**) these are found at 1500 cm^−1^ and ~421 cm^−1^, consistently. FuranCurc-Cu (**8**) shows bands at 1503 cm^−1^ and 418 cm^−1^, together with bands at 1502 cm^−1^ and 484 cm^−1^ in the ThiopheneCurc-Cu (**9**) spectrum (Table 1) [17].

### 2.2. NMR Spectra

The ^1^H NMR spectrum of the ligand N-methyl-pyrCurc (**1**) shows one singlet for the OH proton at 16.66 ppm (due to the strong intramolecular hydrogen bond) and one singlet for the methine vinylic proton at ~5.96 ppm. Protons α and β to the diketone group appear at 6.49 ppm and 7.52 ppm respectively, with a trans coupling constant of 15.5 Hz. Methyl protons appear as singlets at 3.72 ppm (see Table 2). The ^1^H NMR spectrum of FuranCurc (**2**) shows one singlet for the OH proton at 16.06 ppm and a singlet for the methine proton at 6.20 ppm; both protons are involved in a strong intramolecular hydrogen bond (enol tautomer). Protons α to the diketone appear at 6.57 ppm and protons β to the diketone at 7.46 ppm, with a trans coupling constant of ca. 15.7 Hz. Protons α and β to the diketone group appear at 6.57 and 7.46 ppm respectively. The ^1^H NMR spectrum of ThiopheneCurc (**3**) shows a singlet (broad signal) for the OH proton at 16.07 ppm and a singlet for the methine proton at 6.19 ppm. Unsaturated protons α to the diketone group appear at 6.57 ppm and the corresponding β protons at 7.82 ppm, with a trans coupling constant of 15.6 Hz. Aromatic protons were assigned through 1D and 2D experiments. All magnesium complexes maintain the signals previously described except for the enol (-OH) proton signals. The rest of the signals are only affected by the presence of a magnesium nucleus. This effect is observed more in the methine proton of each curcumin analog, observing the following differences in the chemical shifts: 0.53 ppm for N-methyl-pyrCurc-Mg (**4**), 0.7 ppm for FuranCurc-Mg (**5**), and 0.61 ppm for ThiopheneCurc-Mg (**6**) (Table 2) [12,18]. The signals attributed to the α and β protons were not as affected as those from the methine nuclei, showing a difference in rank of 0.13 to 0.06 ppm for the α protons and 0.28 to 0.16 for those corresponding to the β nuclei, although the vinylic protons maintained a J coupling constant of ~16 Hz observed in all cases. Aromatic protons were also assigned using 1D and 2D experiments.

### 2.3. EPR Spectra

The EPR spectra of ligands show diamagnetic spectra, while the EPR spectra of copper complexes of ligands **7**–**9** show a typical four-lines pattern (see Figure 2). The g‖, g_┴_, A‖, and A_┴_ values were obtained directly from the EPR spectra. The g‖and g_┴_ values of complexes **7**–**9** were ca. 2.30 and 2.06, resulting from unpaired electrons in the d_x2-y2_ molecular orbital [17]. The values of g‖ greater than 2.3 suggest an ionic environment for the complexes. The A‖ values ca. 142 × 10^−4^ cm^−1^ are consistent with a typical monomeric distorted square planar geometry. The quotient g‖/ A‖ provides an index of departure from the tetrahedral structure. The quotient values that fall in the range 105–135 cm^−1^ suggest a regular square planar structure. Although the observed values (139–146 cm^−1^) are far from this range, compounds **7** and **8** would be square planar as observed in the crystal structures of compound **8** (see Table 3) [13,17,19,20,21]. 

### 2.4. Single Crystal X-ray Diffraction

The molecular structures of the ligands **1**–**3** and their magnesium (**4**,**6**) and copper (**8**,**9**) complexes were determined by single crystal X-ray diffraction. The crystal data and structure refinements are given in Table 4 and their diagrams are given in Figure 3 and Figure 4.

Ligand **1** exists in two polymorphic forms, both in the monoclinic system and with the same space group *P* 21/c, but not as isomorphs. However, both molecules display a significant similarity (RMSD = 0.0764) (Appendix A).

Figure 1 shows the canonical structures for the corresponding tautomeric representation of ligands **1** to **3**. Conformational searches of the canonical forms **1**–**3** were carried out using MMFF94 implemented in Conflex 7 [22,23]. Appendix A illustrates the nine theoretical extreme conformers of the heterocyclic curcuminoids. Each structure displays a distinctive set of s-*cis* and s-*trans* for the enone (enol) conformation and/or *syn*-*anti* orientation of the heteroatom (N, O, S) relative to the 2,4-pentanedione, thus distinguishing the different families. The corresponding conformer clusters for the ligands 1–3 are shown in Appendix A.

Ligands **1** and **2** display in the solid state the predicted minimum energy conformers with the s-*cis*, s-*cis* enone configuration, which gives them a fully extended nearly planar conformation. Nevertheless, they differ in the orientation of the heterocyclic rings corresponding to *syn*, *syn* for the N-methyl-pyrCurc (**1**), while the *anti-relationship* is observed for FuranCurc (**2**). On the contrary, ligand ThiopheneCurc (**3**) displays in solid-state a disordered structure (0.642:0.358) with s-*trans*, s-*cis*, *anti*, *syn* conformation, which is the less populated conformer (see Figure 3). 

In magnesium complexes **4** and **6**, each of the ligand molecules coordinates to the Mg(II) center as a bidentate (O, O) chelating donor in the meridional style and then acquires a distorted octahedral geometry about the metal ion with DMF solvent molecules in the apical positions. Notably, in complex **6**, the disordered ThiopheneCurc ligand changes from a s-*trans*, s-*cis*, *anti*, *syn* conformation to both s-*cis*, s-*cis, anti*, *anti* (major) and s-*cis*, s-*cis, syn*, *anti* (minor) conformations upon coordination with Mg(II) as depicted in Appendix A.

Upon complexation with copper(II) acetate, ligands **2** and **3** yield the corresponding complexes **8** and **9** respectively. Complex **8** displays a nearly perfect square planar geometry while complex **9** has a square-based pyramidal geometry due to coordination with a DMF molecule with τ**_5_** = 0.14 (Figure 4) [24]. The basal plane of ThiopheneCurc-Cu (**9**) contains two ThiopheneCurc ligands A and B with different conformations. Ligand A (O1-C17) is s-*cis*, s-*cis, anti*, *anti*, whereas ligand B (O21-C37) is s-*cis*, s-*trans*, *anti*, *anti*, Figure 4. Although conformational polymorphism is a well-studied subject, examples of two equal ligands bound to the same metal center that possess different conformations are not documented. Therefore, to the best of our knowledge, we consider the present case the first “conformationally-heteroleptic” complex. Interestingly, the near in-plane geometry involving the β-diketone system is observed only for complex FuranCurc-Cu (**8**), closely resembling the geometry previously found in several curcuminoid copper complexes. [13].

### 2.5. Analysis of Non-Covalent Interactions

The molecular structures of polymorphs **1a** and **1b** contain C-H···π and C-H···O interactions between the N-methyl group with one oxygen of the β-diketone group and another pyrrole ring rendering zigzag layers (Figure 5). Both polymorphs have similar interactions with the same crystalline arrangement. Non-covalent interactions of a diverse nature define attractive arrangements with the geometric parameters of the interactions listed in Table 5. The crystallographic packings are represented in Figure 6 where compound **2** resembles a brick wall. The crystalline arrangement of compound **3** corresponds to a six-sided star along axis c (Figure 7a) with the C3 symmetry supported by three C-H···O and three C-H···S interactions, with donor-acceptor distances of 3.373 and 3.798 Å, respectively (Figure 7b). The packing of the almost planar compound **8** has a perpendicular arrangement of molecules. At the same time, **9** resembles a roof tile supported by three non-classical C-H···O=C interactions between the coordinated carbonyl and methyl groups of DMF with donor-acceptor distances of 3.456, 3.672, 3.750, 3.626 Å (Figure 8). All intermolecular distances and angles were within the expected range reported previously [25,26].

### 2.6. Hirshfeld Surfaces

The Hirshfeld Surface (HS) [27] was plotted in 3D and fingerprint graphs [28] in 2D using CrystalExplorer 21.5 software [29], along with the percentages of contact contributions for the compounds shown in Table 6 and Figure 9. The calculated Hirshfeld surface was mapped over the normalized contact distance d_norm_ (Figure 10) considering the complete fragment of a molecule, including the disordered part for compounds **3**, **4**, **6**, **8**, and **9.** The two molecules of the asymmetric unit of complex **9** were considered, and a close contact for the solvent was taken into account in compounds **4** and **9.** The bright red spots are near the carbonyl groups and the heterocyclic substituent. In compound **8**, such spot is found in the alkene portion, demonstrating a π interaction over this site. Fingerprints were obtained based on HS and percentage of the most relevant contacts: those of type H···H exceed 40% with exception of compound **9** (30.2%). The O···H/H···O contacts fall between 11 and 20% with characteristic wings that can be appreciated in the graphs. The contacts with the lowest percentages were those of type N···H/H···N or S···H/H···S. 

### 2.7. Inhibition of Lipoperoxidation (LP) in Rat Brain Homogenate

In general, the metal complexes of the curcumin analogs did not show antioxidant activity on the lipid peroxidation in a rat brain homogenate model (see Table 7). The IC_50_ value of complex **5** shows lower antioxidant activity than α-tocopherol and BHT. Nevertheless, a significant increase is observed when the FuranCurc (**2**) ligand is compared with its magnesium complex **5**, which triples its antioxidant activity (Table 8) [30].

### 2.8. Cytotoxic Activity

N-methyl-pyrCurc (**1**) is the only ligand possessing high levels of cytotoxicity, a noteworthy fact since the phenolic groups to whom this property is conferred have not been found [31] (see Table 9). In general, a significant increase in the cytotoxic activity of the magnesium and copper complexes **4**–**9** with respect to their free ligands is observed. On the contrary, the N-methyl-pyrCurc-Mg (**4**) complex showed poor cytotoxic activity, just the opposite to the N-methyl-pyrCurc-Cu (**7**), FuranCurc-Mg (**5**), and ThiopheneCurc-Mg (**6**) complexes which exhibited an important cytotoxic effect (see Table 9). Compounds **5**–**6** and **9** showed great selectivity towards certain lines specifically, which has never been observed on these compounds until now (see Table 9).

The results indicate that the FuranCurc-Mg (**5**), N-methyl-pyrCurc-Cu (**7**), and FuranCurc-Cu (**8**) complexes have important cytotoxic effects against the U251, PC-3, K562, and HCT-15 cell lines, the IC_50_ values of three homoleptic complexes being similar to cisplatin in some cases (see Table 10). The FuranCurc-Mg complex showed itself to be almost as cytotoxic as cisplatin in the colon cell line and is also almost three times less toxic than cisplatin in healthy cells. These results in vitro could suggest the good therapeutic potential of this compound.

## 3. Materials and Methods 

All chemicals were available commercially and purchased from Sigma-Aldrich. The solvents were purified by conventional methods prior to use [32]. 

### 3.1. Physical Measurements

The melting points were determined on an Electrothermal Engineering IA9100 × 1 melting point apparatus and are uncorrected. 

### 3.2. Spectroscopic Determinations

The IR absorption spectra were recorded in the range of 4000–230 cm^−1^ as KBr pellets on a BRUKER Tensor 27 spectrophotometer. The ^1^H and ^13^C NMR spectra were recorded in dimethyl sulfoxide (DMSO-*d_6_*) on a Bruker Fourier 300 MHz and Varian Unity Inova 500 MHz spectrometer using TMS as an internal reference. The EPR spectra were recorded in DMF at liquid nitrogen temperature (77 K) on an Electron Paramagnetic Resonance Spectrometer JEOL, JES-TE300, ITC Cryogenic System, Oxford. Mass spectra were recorded in a JEOL, SX 102 A spectrometer on Bruker Microflex equipped with MALDI-Flight time. The single-crystal X-ray diffractions (SCXRD) were obtained in a Bruker diffractometer, model D8 Venture, equipped with Mo radiation (λ = 0.71073Å) and Cu radiation (1.54178Å), a CCD two-dimensional detector, and a low-temperature device. The data collection and data reduction were performed by APEX and SAINT-Plus programs. These structures were solved by direct methods using SHELX-2013 software and refined by the Full-matrix least-squares procedure on F2 using the SHELX-2008 program [33]. 

### 3.3. Inhibition of Lipid Peroxidation on Rat Brain

#### 3.3.1. Animal

Adult male Wistar rats (200–250 g) were provided by the Instituto de Fisiología Celular, Universidad Nacional Autónoma de México (UNAM). The procedures and care of animals were conducted in conformity with the Mexican Official Norm for Animal Care and Handling NOM-062-ZOO-1999. They were maintained at 23 ± 2 °C on a 12/12 h light-dark cycle with ad libitum access to food and water.

#### 3.3.2. Rat Brain Homogenate Preparation

Animal sacrifice was carried out avoiding unnecessary pain. The rats were sacrificed with CO_2_. The cerebral tissue (whole brain) was rapidly dissected and homogenized in phosphate-buffered saline (PBS) solution (0.2 g of KCl, 0.2 g of KH_2_PO_4_, 8 g of NaCl, and 2.16 g of NaHPO_4_._7_H_2_O/L, pH adjusted to 7.4) as described elsewhere [34,35] to produce a 1/10 (*w*/*v*) homogenate. The homogenate was centrifuged at 800 rcf (relative centrifugal field) for 10 min. The supernatant protein content was measured using Folin and Ciocalteu’s phenol reagent [36] and adjusted with PBS at 2.666 mg of protein/mL.

#### 3.3.3. Induction of Lipid Peroxidation and Thiobarbituric Acid Reactive Substances (TBARS) Quantification

As an index of lipid peroxidation, TBARS levels were measured using rat brain homogenates according to the method described by Ng and co-workers [37], with some modifications. A supernatant (375 µL) was added with 50 µL of 20 µM EDTA and 25 µL of each sample concentration dissolved in DMSO (25 µL of DMSO for the control group) and incubated at 37 °C for 30 min. The lipid peroxidation was started adding 50 µL of freshly prepared 100 µM FeSO_4_ solution (final concentration 10 µM) and incubated at 37 °C for 1h. The TBARS measurements were obtained as described by Ohkawa and co-workers [38], with some modifications. To each tube 500 µL of TBA reagent (0.5% 2-thiobarbituric acid in 0.05 N NaOH and 30% trichloroacetic acid, in 1:1 ratio) was added and the final suspension was cooled on ice for 10 min, centrifuged at 13,400 rcf for 5 min and heated at 80 °C in a water bath for 30 min. After cooling at room temperature, the absorbance of 200 µL of supernatant was measured at λ = 540 nm in a Microplate Reader Synergy/HT BIOTEK Instrument, Inc., Winooski, VT, USA. The concentration of TBARS was calculated by interpolation on a standard curve of tetra-methoxypropane (TMP) as a precursor of MDA [38]. Results are expressed as n moles of TBARS per mg of protein. The inhibition ratio (IR [%]) was calculated using the formula IR = (C − E) × 100/C, where C is the control absorbance and E is the sample absorbance. Butylated hydroxytoluene (BHT) and α tocopherol were used as positive standards. All data are presented as mean ± standard error (SEM). The data were analyzed by one-way analysis of variance (ANOVA) followed by Dunnett’s test for comparison against a control. Values of *p* ≤ 0.05 (*) and *p* ≤ 0.01 (**) were considered statistically significant.

### 3.4. Cytotoxic Activity in Human Tumor Cells

The cytotoxicity of all compounds was tested against six cancer cell lines: U251 (human glioblastoma cell line), PC-3 (human Caucasian prostate adenocarcinoma), K562 (human Caucasian chronic myelogenous leukemia), HCT-15 (human colon adenocarcinoma), MCF-7 (human mammary adenocarcinoma), and SKLU-1 (human lung adenocarcinoma). Cell lines were supplied by the U.S. National Cancer Institute (NCI). The cell lines were cultured in an RPMI-1640 medium supplemented with 10% fetal bovine serum, 2 mL L-glutamine, 10,000 units/mL penicillin G sodium, 10,000 µg/mL streptomycin sulfate, 25 µg/mL amphotericin B (Invitrogen/Gibco™, Thermo Fisher Scientific, Waltham, MA, USA), and 1% non-essential amino acids (Gibco). They were maintained at 37 °C in a humidified atmosphere with 5% CO_2_. The viability of the cells used in the experiments exceeded 95% as determined with trypan blue. The human tumor cytotoxicity was determined using the protein-binding dye sulforhodamine B (SRB) in a microculture assay to measure cell growth, as described in the protocols established by the NCI [39,40,41].

### 3.5. Synthesis of Compounds

The general synthetic procedure for N-methyl-pyrCurc **1**, FuranCurc **2**, ThiopheneCurc **3**, N-methyl-pyrCurc-Mg **4**, FuranCurc-Mg **5** and ThiopheneCurc-Mg **6,** N-methyl-pyrCurc-Cu **7**, FuranCurc-Cu **8,** ThiopheneCurc-Cu **9** is shown in Figure 1.

Compound **1**. For N-methyl-pyrCurc, 3.23 mmol of boric acid and 6.46 mmol of acetylacetone were dissolved in ethyl acetate (A-flask). In another flask, 12.93 mmol of N-methyl-2-pyrrolecarboxaldehyde and 15.52 mmol of tributyl borate were mixed (B-flask). Both flasks were stirred and submitted to reflux for 2h. Then, the contents of “A-flask” were poured into “B-flask”, the mixture was stirred for 30 min, with the dropwise addition of 15.52 mmol of n-butylamine (n-butylamine dissolved in 10 mL of EtOAc). The reaction mixture was refluxed under an N_2_ atmosphere for 60 h. Then, 15.52 mmol of n-butylamine was added and refluxing was followed for an additional 40 h. After 100 h, EtAcO: H_2_O extractions (4 × 50mL) were carried out. The organic phase was dried with sodium sulfate, filtered, and concentrated under reduced pressure. The concentrated product was crystallized in CH_2_Cl_2_/MeOH affording N-methyl-pyrCurc as purple crystals, m.p. 159.5–160 °C. Yield: 30%. ^1^H NMR (500 MHz DMSO-*d_6_*): δ 3.72 (s, 6H), 5.96 (s, 1H), 6.15 (dd, 2H_aryl_, J 3.90; 2.46), 6.49 (d, 2H_Vinyl_, J 15.52), 6.79 (dd, 2H_aryl_, J 3.92; 1.68), 7.02 (t, 2H_aryl_, J 2.02; 2.02), 7.52 (d, 1H_Vinyl_, J 15.53), 16.69 (br s, 1H) ppm, ^13^C NMR (^13^C 1H 125 MHz, DMSO-*d_6_*): δ 34.35 (C-H), 101.11 (C-H), 109.36 (C_aryl_), 112.15 (C_aryl_), 119.05 (C-H), 128.44 (C_aryl_), 128.58 (C-H), 130.16 (C_aryl_), 183.11 (C=O) ppm, IR: 3049.18, 1608.99, 1497.82, 959.55 cm^−1^, HRMS: observed: 283.1442; estimated: 282.1368.

Compound **2**. For FuranCurc, 3.23 mmol of boric acid was dissolved in EtOAc to later add to 6.46 mmol of acetylacetone (A-flask). Separately, 12.93 mmol of Furan-2-carboxaldehyde and 15.52 mmol of tributyl borate were mixed (B-flask) and both flasks were stirred with reflux for 2 h; after this time, the contents of “A-flask” were poured into “B-flask”. The mixture was homogenized and 12.8 mmol of n-butylamine was added dropwise (n-butylamine dissolved in 10 mL of EtOAc) and refluxed for 48 h. Then, four EtAcO: H_2_O extractions (4 × 50mL) were performed and the joint organic phase was dried with sodium sulfate, filtered, and concentrated under reduced pressure. The concentrated mixture was dissolved in CHCl_2_ and MeOH was added, affording FuranCurc, orange crystals, m.p. 130.5–131 °C. Yield: 50%. ^1^H NMR (500 MHz DMSO-*d_6_*): δ 6.20 (s, 1H), 6.57 (d, 2H_Vinil_, J 15.72), 6.66 (dd, 2H_aryl_, J 3.44; 1.79), 6.97 (d, 5H_aryl_, J 3.40), 7.46 (d, 2H _Vinyl_, 15.72), 7.88 (s, 2H_aryl_), 16.06 (br s, 1H), ppm, ^13^C NMR (^13^C {^1^H} 125 MHz, DMSO-*d_6_*): δ 101.95 (C-H), 113.03 (C_aryl_), 116.10 (C_aryl_), 121.18 (C-H), 126.92 (C-H), 146.08 (C_aryl_), 151.01 (C_aryl_), 182.46 (C=O) ppm, IR: 3124.24, 1623.33, 1486.88, 957.31 cm^−1^, HRMS: observed: 257.0808; estimated: 256.0736.

Compound **3**. For ThiopheneCurc, 3.23 mmol of boric acid and 6.46 mmol of acetylacetone were dissolved in EtOAc (A-flask). In another flask, 12.93 mmol of 2-Thiophenecarboxaldehyde and 15.52 mmol of tributyl borate were mixed (B-flask). Both flasks were stirred and submitted to reflux for 2h, and the contents of “A-flask” were poured into “B-flask”. The mixture was homogenized, 12.8 mmol of n-butylamine dissolved in 10 mL of EtOAc was added dropwise, and the mixture was refluxed for 48 h. Then, four EtAcO: H_2_O extractions (4 × 50mL) were carried out, and the joint organic phase was dried with sodium sulfate, filtered, and concentrated under reduced pressure. Crystallization in CH_2_Cl_2_/MeOH afforded ThiopheneCurc, orange crystals, m.p. 183.5–184 °C. Yield 55%. ^1^H NMR(500 MHz DMSO-*d_6_*): δ 6.19 (s, 1H), 6.57 (d, 2H_Vinil_, J 15.61), 7.18 (dd, 2H_aryl_, J 5.04; 3.59), 7.54 (d, 2H_aryl_, J 3.10), 7.75 (d, 2H_aryl_, J 5.06), 7.82 (d, 2H _Vinyl_, J 15.63), 16.07 (br s, 1H), ppm, ^13^C NMR (^13^C {^1^H}) 125 MHz, DMSO-*d_6_*): δ 101.52 (C-H), 122.70 (C-H), 128.75 (C_aryl_), 130.05 (C_aryl_), 132.05 (C_aryl_), 133.23 (C-H), 139.83 (C_aryl_), 182.52 (C=O) ppm, IR: 3049.18, 1608.99, 1497.82, 959.55 cm^−1^, HRMS: observed: 289.0351; estimated: 288.0279.

Compound **4**. For N-methyl-pyrCurc-Mg, in a round bottom flask, 1 mmol of N-methyl-pyrCurc was dissolved in 15 mL of EtAcO and 0.6 mmol of magnesium(II) acetate dissolved in MeOH was added dropwise at room temperature. The mixture was stirred for 24h, and the resulting fine powder was filtered off in vacuo and washed with H_2_O, EtAcO, and Et_2_O. m.p. 208.3–208.7 °C. Yield 80%. ^1^H NMR (500 MHz DMSO-*d_6_*): δ 3.65 (s, 6H), 5.43 (s, 1H), 6.06 (dd, 2H_aryl_, J 3.83; 2.56), 6.42 (d, 2H _Vinyl_, J 15.39), 6.56 (dd, 2H_aryl_, J 3.82; 1.63), 6.88 (s, 2H_aryl_), 7.36 (d, 2H _Vinyl_, J 15.33) ppm, ^13^C NMR (^13^C {^1^H} 125 MHz, DMSO-*d_6_*): δ 34.07 (C-H), 103.19 (C-H), 108.72 (C_aryl_), 109.83 (C_aryl_), 124.70 (C-H), 126.14 (C_aryl_), 126.19 (C-H), 130.63 (C_aryl_), 181.51 (C=O) ppm, IR: 1536.66, 1508.85, 962.27, 445.62 cm^−1^, HRMS: observed: 587.2510; estimated: 586.2430.

Compound **5**. For FuranCurc-Mg, 1 mmol of FuranCurc was dissolved in 15 mL of EtAcO and 0.6 mmol of magnesium(II) acetate dissolved in MeOH was added dropwise at room temperature. The mixture was stirred for 24 h. The fine powder formed was filtered off in vacuo and washed with H_2_O, EtAcO, and Et_2_O. m.p. 220–220.5 °C. Yield 80%. ^1^H NMR (500 MHz DMSO-*d_6_*): δ 5.55 (s, 1H), 6.51 (d, 2H_Vinil_, J 15.51), 6.58 (dd, 2H_aryl_, J 3.36; 1.84), 6.76 (d, 2H_aryl_, J 3.40), 7.19 (d, 2H _Vinyl_, 15.51), 7.75 (d, 2H_aryl_, J 1.78) ppm, ^13^C NMR (^13^C ^1^H 125 MHz, DMSO-*d_6_*): δ 104.19 (C-H), 112.93 (C_aryl_), 113.22 (C_aryl_), 124.03 (C-H), 128.40 (C-H), 144.87 (C_aryl_), 152.38 (C_aryl_), 181.26 (C=O) ppm, IR: 1519.75, 1420.18, 958.60, 499.46 cm^−1^, HRMS: observed: 535.1248; estimated: 534.1165

Compound **6**. For ThiopheneCurc-Mg, 1 mmol of ThiopheneCurc was dissolved in 15 mL of EtAcO and 0.6 mmol of magnesium(II) acetate dissolved in MeOH was added dropwise at room temperature with stirring for 24h. Then, the fine powder formed was filtered off in vacuo and washed with H_2_O, EtAcO, and Et_2_O. m.p. 286.2–286.5 °C. Yield 85%. ^1^H NMR (500 MHz DMSO-*d_6_*): δ 5.58 (s, 1H), 6.48 (d, 2H_Vinil_, J 15.30), 7.10 (t, 2H_aryl_, J 4.34), 7.36 (d, 2H_aryl_, J 3.55), 7.54 (d, 2H _Vinyl_, J 15.05), 7.58 (d, 2H_aryl_, J 1.16) ppm, ^13^C NMR (^13^C {^1^H} 125 MHz, DMSO-*d_6_*): δ 103.63 (C-H), 127.80 (C-H), 128.79 (C_aryl_), 129.40 (C_aryl_), 129.92 (C_aryl_), 130.16 (C-H), 141.48 (C_aryl_), 180.44 (C=O) ppm, IR: 1521.26, 1497.10, 956.18, 475.63 cm^−1^, HRMS: observed: 599.0331; estimated: 598.0251.

Compound **7**. For N-methyl-pyrCurc-Cu, 1 mmol of N-methyl-pyrCurc was dissolved in 15 mL of EtAcO and 0.6 mmol of copper(II) acetate dissolved in MeOH was added dropwise at room temperature. The mixture was stirred for 24 h. The fine powder was filtered off in vacuo and washed with H_2_O, EtAcO, and Et_2_O. m.p. 237.2–237.6 °C. Yield: 50%. EPR: g‖ 2.30241, g⊥ 2.06319, A‖ 15.918, A⊥ 1.144. IR: 1511.14, 977.16, 489.16 cm^−1^. HRMS: observed: 626.1949; estimated: 625.1876.

Compound **8**. For FuranCurc-Cu, 1 mmol of FuranCurc was dissolved in 15 mL of EtAcO and 0.6 mmol of copper(II) acetate dissolved in MeOH was added dropwise at room temperature with stirring for 24 h. The fine powder formed was filtered off and washed with H_2_O, EtAcO, and Et_2_O. m.p. 225.5–226 °C. Yield: 93%, EPR: g‖ 2.29660, g⊥ 2.06161, A‖ 16.203, A⊥ 1.334. IR: 1511.14, 957.31 cm^−1^ 479.14 cm^−1^, HRMS: observed: 574.0683; estimated: 573.0611.

Compound **9**. For ThiopheneCurc-Cu, 1 mmol of ThiopheneCurc was dissolved in 15 mL of EtAcO and 0.6 mmol of copper(II) acetate dissolved in MeOH was added dropwise at room temperature with stirring for 24 h. The fine powder formed was filtered off and washed with H_2_O, EtAcO, and Et_2_O. m.p. 275–275.5 °C. Yield: 95%, EPR: g∥ 2.30241, g⊥ 2.06319, A∥ 15.918, A⊥ 1.144. IR: 1502.47, 952.83, 484.91 cm^−1^. HRMS: observed: 637.9770; estimated: 636.9697.

## 4. Conclusions

The synthesis of six new homoleptic complexes with magnesium and copper was achieved with three different curcumin analog ligands (compound **1** has two polymorphs), and their crystal structures reveal square planar and square-based pyramid geometry for the copper complexes and octahedral geometry for the magnesium complexes. Therefore, the use of DMSO and DMF as solvents for crystallization is important in the formation of single crystals. The finding of different conformations for the same ligand bound to a single metal center, as occurs in thiopheneCurc-Cu, is of capital importance since it becomes the first reported case described in the literature to the best of our knowledge. The supramolecular description of ThiopheneCurc constitutes a rare case of an arrangement that led to the apparent six-peak star arrangement of two overlaid trimeric structures. The remarkable cytotoxic effect found for FuranCurc-Mg, N-methyl-pyrCurc-Cu, and FuranCurc-Cu against U251, PC-3, K562, and HCT-15 human cancer cell lines surprisingly equated with or surpassed that of cisplatin. Among them, the FuranCurc-Mg complex stands out because it is almost three times less toxic than cisplatin in the healthy cell line (COS7), so it would have great potential as a therapeutic agent.

## Data Availability

Not applicable.

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
