# Peer review of "Homoleptic Complexes of Heterocyclic Curcuminoids with Mg(II) and Cu(II): First Conformationally Heteroleptic Case, Crystal Structures, and Biological Propertiesâ€"

_molecules, 2023, doi:10.3390/molecules28031434_

Round 1
Reviewer 1 Report
The presented manuscript entitled “Homoleptic complexes of heterocyclic curcuminoids with Mg and Cu: first conformationally heteroleptic case, crystal structures and biological properties”, by Meza-Morales et al., describes synthesis, chemical characterization and anticancer in vitro evaluation of nine new compounds (three ligands and six metal complexes). The manuscript is well written, and the topic is important and show significant movement forward in the field of antitumor activity of Mg and Cu(II) complexes. Crystal structures of synthesized complexes showed interesting geometry of these complexes, and definitely confirmed proposed structures. Biological results are interesting too with antitumor activity similar to cisplatin for one magnesium and copper compound. Parallel to this activity, these complexes express lower citotoxicity against health cells than cisplatin. In order to improve the quality of presented manuscript, I suggest only few minor corrections:
1. In the part Results and discussion of NMR spectra, it should be also mentioned that in the NMR spectra of Mg complexes signal for OH group is missing due to ligand coordination to Mg ion. If it is possible, the same thing should be commended in the IR spectra.
Also some typo mistakes should be corrected:
1. In the Abstract section Infrared spectroscopy abbreviation should be corrected to IR instead of IS.
2. Through the whole manuscript name of metal salt precursor should be “metal(II) acetate” not “metal acetate (II)” as it was written.
3. In the section Materials and methods, Synthesis part, authors should correct 1H RMN and 13C RMN to 1H NMR and 13C NMR.
4. Finally, DMSO-d6 should be uniformed through the text either DMSO-d6 or DMSO-d6.
5. On the page 2, IC50 should be IC50 with subscript, and also at page 4 dx2-y2 with subscript as well as 10-4 and cm-1 with superscript.
In my opinion, presented manuscript is very well written, the results are well documented, and in my opinion manuscript can be accepted for publishing in the Molecules after minor corrections.
Reviewer 2 Report
In my opinion the manuscript entitled “Homoleptic complexes of heterocyclic curcuminoids with Mg and Cu: first conformationally heteroleptic case, crystal structures and biological properties” is consistent with the subject of the journal and the work is up-to-date. The introduction is short but provides a good background for the issues presented. The authors describe the interesting research of copper and magnesium complexes of heterocyclic curcuminoid. The synthesis and crystallization process is quite good describe. The chemical characterization of obtained compounds is very deep and gives interesting results. Moreover the biological evaluation gives great results what makes this manuscript valuable multidisciplinary report. Unfortunately, the authors did not manage to avoid a few mistakes, which, however, are classified as minor and do not diminish work in any way. Nevertheless, I think that this work requires minor revision before publication in Molecules. The most important matter needed correction are listed, below:
1. The short and most common use name of Infrared Spectroscopy is IR. I recommend to use this short instead IS.
2. It will be nice to add the simulation of EPR spectra, which can provide more EPR parameters.
3. There are several editorial bugs, punctuation errors and also typos in the work. Before the publication all of them should be improve throughout. For example:
- to many or missing ‘space’, the bold font in text line 45, 49.
- The short of for example is ‘e.g.’ not ‘e.gr.:’.
- The end of sentence should be end by dot not semicolon (line 69).
- Each of band absorption band should have a unit (cm-1), please correct at line 80.
- At line 104 ‘weren’t’ should be written “were not”.
- The molecular orbital ‘dx2-y2’ please use index (line 114). At lines 116, 118, 119 please use upper index.
- The captions of figures and tables should have the same formatting throughout the work, please standardize it.
- Please complete the valence or the degree of oxidation in the symbols 'Mg' . I also recommend re-releasing the title in this respect.
- Line 180 to many dots.
4. In Material and methods please indicate the origin and possible purification of all chemical reagents used in the tests.
5. Please remove the section which is not mandatory and the end of manuscript ‘Patent’.
6. In Supplementary Information Figure S52-54 are low quality and should be loaded with better resolution. Also the UV-Vis spectra (Figure S58-66) have to be in better resolution and without the handmade note.
7. Moreover there is a one more point which could positively influence the reception of work and make it more valuable. It is not require, but only recommended to expand the paper, namely adding graphical abstract.
